# High-Voltage Electrostatic Fields Adversely Affect the Performance of Diamondback Moths over Five Consecutive Generations

**Li Jia, Shicai Xu, Huanzhang Shang, Jiao Guo, Xia Yan, Changhai Liu, Guangwei Li and Kun Luo ***

Shaanxi Engineering and Technological Research Center for Conversation and Utilization of Regional Biological Resources, College of Life Science, Yan'an University, Yan'an 716000, China
* Correspondence: luok1985@gmail.com

**Abstract:** Changing electrical environments can influence the performance of herbivorous insects and adversely affect their control strategies. The diamondback moth, *Plutella xylostella* (L.), is a pest that devastates cruciferous vegetables. An age–stage, two-sex life table of *P. xylostella* over multiple generations was established to describe the effect of varying high-voltage electrostatic field (HVEF) exposure on their performance after the age-cohort eggs were exposed to HVEF at an intensity of 5.0 kV/cm for different durations. The results show that direct HVEF exposure adversely affected the population dynamics parameters of *P. xylostella* over multiple generations. In particular, the net reproduction rate, intrinsic natural increase rate, and finite increase rate of the *P. xylostella* population significantly decreased in the third and fifth generations under HVEF exposure for 10 min, while the mean generation time and doubling time significantly increased. Similarly, HVEF exposure for 10 min rapidly reduced the survival rate of adult *P. xylostella* in the first generation, and subsequently, it declined evenly and slowly. Meanwhile, the fecundity parameters of *P. xylostella* revealed that HVEF exposure for 10 min had the strongest inhibition effect on reproduction over five consecutive generations. In addition, HVEF exposure significantly increased the superoxide dismutase activity to produce extra hydrogen peroxide; however, increased catalase and peroxidase activity or reduced peroxidase activity triggered the accumulation of malondialdehyde in instar *P. xylostella*, especially after 10 min of treatment. The present findings provide experimental evidence and a theoretical basis for developing control strategies for *P. xylostella* under new HVEF environments.

**Keywords:** high-voltage electrostatic field (HVEF); *Plutella xylostella*; Cruciferae vegetables; two-sex life table; population dynamics

## 1. Introduction

The diamondback moth, *Plutella xylostella* Linnaeus (Lepidoptera: Plutellidae), is a pest that devastates cruciferous vegetables such as pakchoi *Brassica chinensis* Linnaeus [1–3]. During the larval stage, especially the third and fourth instars, larvae gnaw on fresh leaves, severely damaging the leaves and causing significant yield losses in cruciferous vegetable production [4]. To effectively suppress the damage and keep it below the economic threshold, chemical spraying is still a crucial measure for managing the population dynamics of *P. xylostella* in agricultural production [5]. However, in recent decades, the economically irrational frequent spraying of several kinds of pesticides has significantly accelerated the development of pests' resistance to diverse chemicals [6,7]. It has been demonstrated that *P. xylostella* has already developed resistance to approximately 50 kinds of insecticides [8–11]. This is likely because its genetic plasticity stimulates an accumulating increase in resistance to such chemicals, which would serve as a genetic basis for it to adapt to the altered environment, and this poses a serious challenge to cruciferous vegetable production worldwide.

In addition, the artificial electric fields derived from modern industrial civilization have greatly increased the intensity of the natural electric fields to which organisms are directly exposed [12–14]. Previous studies have demonstrated that dramatic alterations in the electric environment affect the performance of plants and animals [15,16]. Similarly, insects are extremely sensitive to environmental changes, and can develop certain adaptive strategies owing to their high evolutionary rate, short growth cycle, small body size, and poor migration of larvae [17]. For instance, direct exposure of the cereal aphid *Sitobion avenae* Fabricius (Hemiptera: Aphididae), a prevalent and economically important wheat pest worldwide, to a high-voltage electrostatic field (HVEF) for 20 min at an intensity of 4 kV/cm had strong adverse effects on the population dynamics parameters of the aphids. However, using the same treatment on *S. avenae* over multiple generations, it was revealed that the aphids gradually recovered from the adverse effects of direct HVEF exposure over the generations, suggesting that the aphids' bodies changed in response to stress [18–20]. Moreover, when *S. avenae* were directly exposed to and fed on plant seeds exposed to the same intensity of HVEF, their antioxidative enzyme activity was affected, which supports the results of previous studies [21]. Similar findings were reported in many other insects, such as *Bombyx mori* (Lepidoptera: Bombycidae) [22], *Drosophila melanogaster* (Diptera: Drosophilidae) [23], and *Myzus persicae* (Hemiptera: Aphididae) [18,24]. Therefore, the rapid adaptation of insects, especially species with high genetic plasticity, to novel electrical environment alternations could make it more difficult to develop environmentally friendly pest control strategies.

To develop an alternative method that uses a more effective and sustainable pest management strategy to control *P. xylostella* larvae, we determined the effects of direct exposure to HVEF on *P. xylostella* larvae. In the field, *P. xylostella* typically passes through multiple generations during the plant growth period. To simulate high-voltage HVEF stress, we directly exposed newborn *P. xylostella* eggs (within 24 h after birth) to an HVEF field for five consecutive generations. An age–stage, two-sex life table of *P. xylostella* was established to determine the effects of direct HVEF exposure on the growth, development, and reproduction of *P. xylostella* over multiple generations. Meanwhile, the antioxidative enzyme activity and malondialdehyde (MDA) levels of fourth-instar *P. xylostella* were evaluated to characterize the physiological alterations after HVEF treatment. The results of the current study have increased our understanding of the performance of *P. xylostella* under HVEF stress, and have provided experimental data and a theoretical basis for the development of *P. xylostella* control strategies in novel HVEF environments.

## 2. Materials and Methods

### 2.1. Insect Specimens and Rearing Conditions

Larvae of *P. xylostella* were collected from a vegetable greenhouse in Yan'an (109°35′ E, 36°63′ N), Shaanxi Province, China. The larvae were taken to the Insect Physiology and Ecology Laboratory of Yan'an University. After five generations of stable reproduction with an artificial diet, they were used as test specimens. In order to avoid biotic and abiotic effects on the *P. xylostella* population, they were maintained on an artificial diet and placed in an artificial climate chamber with a constant temperature of 25 ± 1 °C, relative humidity of 55 ± 10%, and a photoperiod of 12:12 (L:D) until pupation. The preparation of the artificial diet and its main components and proportions was carried out according to the China Invention Patent (Publication No. CN103478486A, 1 January 2014). To attract the moth larvae, the linseed oil in the formulation was exchanged for canola oil [25]. A newly prepared artificial diet was exchanged with the old one every two days. To produce eggs, the pupae of *P. xylostella* were transferred to oviposition cages for adult emergence, and the adults were fed with 10% honey solution until egg laying and death. The cage was covered with a black cloth and maintained in the growth chamber. Newborn larvae of *P. xylostella* were reared in the same manner until a sufficient number was obtained for experiments. To obtain the age-cohort eggs of *P. xylostella* for HVEF treatment, a multitude of male and female diamondback moths were transferred to the oviposition cage, and a

piece of spawning paper was suspended in the middle of the cage. After 24 h, the spawning paper with the age-cohort eggs of *P. xylostella* was removed from the oviposition cage. These newly laid eggs were used in the following experiments within 24 h.

### 2.2. HVEF Treatment

The HVEF generator used in this study (WJ-II, 0–100 kV output voltage) was purchased from Wuxi Boya Electronic Technology Co., Jiangsu Province, China. Two parallel rectangular aluminum plates (area, $50 \times 50$ cm$^2$; distance between two plates, 8.0 cm) were installed in a wooden frame to form an electrical field. The output wires from the HVEF generator were connected to the aluminum plates, while a ground wire was connected to the field to avoid electrostatic damage. For each treatment, 100 newly laid eggs of *P. xylostella* (within 24 h after birth) were collected and placed in uncovered Petri dishes, which were directly exposed to HVEF with an intensity of 5.0 kV/cm (as determined prior to the experiment, this treatment intensity was appropriate) for 5, 10, 15, and 20 min. Another 100 newborn eggs without HVEF exposure were used as controls (Figure 1). There were three biological replicates for each treatment.

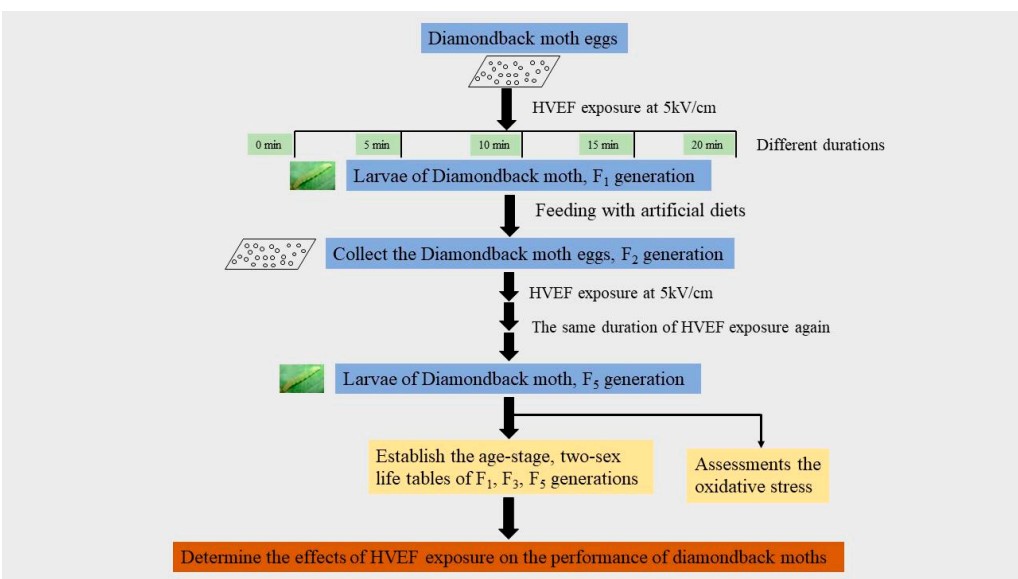

**Figure 1.** Experimental design for investigating suppression of *P. xylostella* performance over five generations, in response to direct exposure of eggs to HVEF at 5 kV/cm for different durations. The blue rectangles represent the different stages and generations of *P. xylostella* concerned in this study. $F_1$ to $F_5$ generation represent the first to the fifth generation of *P. xylostella.* The golden yellow rectangles represent the methods that employed into this study. The orange rectangles represent the main purpose of this study.

### 2.3. Life Table Analysis

The following experiments were conducted in controlled greenhouse conditions, as described previously. After HVEF treatment, the eggs were individually placed in feeder boxes and transferred to feeder cups (50 mL) containing feed for single rearing after they hatched. A freshly prepared artificial diet was exchanged with the old one every 1 to 2 days during the experimental period. Daily individual observations of larval mortality, larval molt, pupation time, and adult emergence were recorded. The survival rate and development time were obtained for all stages. When the larvae pupated, the artificial diet was removed to keep the inside of the bearing cups dry. When the pupae became adults, males and females that emerged on the same day in each treatment group were paired, and each pair was placed in a plastic oviposition container (14 mL). Each oviposition container also included a small cotton ball soaked in a 10% honey solution on which the adults were able to to feed, and a $3 \times 4$ cm$^2$ rectangular piece of spawning paper

was hung inside the cup to collect the eggs. The number of newborn eggs was recorded, and the eggs were removed daily until all female diamondback moths died.

To determine the effects of HVEF on multiple generations of *P. xylostella*, the newly laid eggs produced by the adults in each test group were collected within 24 h for five consecutive generations. They were directly exposed to HVEF for the same treatment time using the physical method described previously. From the onset of reproduction for each female, the number of newborn eggs was recorded daily, and the life data and parameters of the first ($F_1$), third ($F_3$), and fifth ($F_5$) generations of *P. xylostella* were analyzed in the present study. The life table parameters of the population growth of *P. xylostella*, including net reproductive rate ($R_0$), intrinsic natural increase rate ($r$), mean generation time ($T$), finite increase rate ($\lambda$), and doubling time ($dt$), were evaluated to determine the effects of direct HVEF exposure on performance. The life table parameters for each moth cohort were calculated using the equations below [26]. The stage-specific survival rate ($l_x$) describes the probability of survival of the individuals in a given population under treatment, and the stage-specific fecundity of the total population ($m_x$) reflects the average number of offspring born to each individual. The parameters $l_x$ and $m_x$ were calculated as follows:

$$l_x = \sum_{j=1}^{m} s_{xj} \tag{1}$$

$$m_x = (\sum_{j=1}^{m} s_{xj} f_{xj}) / \sum_{j=1}^{m} s_{xj} \tag{2}$$

where $S_{xj}$ is the age- and stage-specific survival, including both the survival situation and the stage differentiation, and $f_{xj}$ is the age- and stage-specific fecundity.

The net reproductive rate was defined as the age-specific survival rate and fecundity for each individual during its lifetime, including females, males, and individuals that died in immature stages, using the following equation:

$$R_0 = \sum_{x=0}^{\infty} \sum_{j=1}^{m} s_{xj} f_{xj} = \sum_{x=0}^{\infty} l_x m_x \tag{3}$$

In addition, the parameter $r$ was considered to describe the maximum instantaneous growth rate of the population under stable conditions, and was calculated as shown in Equation (4):

$$\sum_{x=0}^{\infty} e^{-r(x+1)} l_x m_x = 1 \tag{4}$$

Meanwhile, other parameters of the population dynamics, $T$, $\lambda$, and $dt$, were calculated based on the above two parameters; $T$ describes the time required for the population to develop for a whole generation, and was calculated using Equation (5). Parameters $\lambda$ and $dt$ represent the population growth relative to the population size, and were calculated using Equations (6) and (7):

$$T = (\ln R_0)/r \tag{5}$$

$$\lambda = e^r \tag{6}$$

$$d_t = \ln 2/r \tag{7}$$

Furthermore, to estimate the effects of direct HVEF exposure on *P. xylostella* reproduction, the relevant data from the life tables were employed to calculate the fecundity parameters, including the adult preoviposition period (APOP), oviposition period (OP), oviposition day (OD), and eggs per day during the oviposition period.

### 2.4. Oxidative Stress Assessment

To determine whether direct HVEF exposure affected physiological alterations in herbivores, the antioxidative enzyme activity and malondialdehyde (MDA) levels of fourth-instar *P. xylostella* were evaluated for each treatment. In parallel with the life table data collection, when eggs reached the fourth instar within 24 h after HVEF stress, the samples from the first and third generations were collected individually in 1.5 mL centrifuge tubes (6 fourth-instar larvae per tube). Then, the collected samples were snap-frozen using liquid nitrogen and stored in an ultra-low-temperature refrigerator at −80 °C for enzyme activity assessment. The superoxide dismutase (SOD), catalase (CAT), and peroxidase (POD) activity, as well as the MDA level of fourth-instar *P. xylostella,* were determined according to commercial assay kits purchased from Nanjing Jiancheng Bioengineering Institute (SOD: item no. A001-3-2; CAT: A007-2-1, POD: A084-3-1, MDA: A003-1-2). The reaction mixture of fourth-instar *P. xylostella* tissue was prepared according to the reference manuals, and the absorbance of the mixture of all samples was determined within 10 min. The antioxidative enzyme activity in *P. xylostella* was expressed as U/mgprot, and the MDA level was expressed as nmol/mgprot. Three biological replicates were performed per treatment.

### 2.5. Statistical Analysis

The population parameters for all *P. xylostella* individuals in the study were analyzed according to an age–stage, two-sex life table using TWOSEX-MSChart software [27]. Based on the TWOSEX-MSChart software, the bootstrap technique with 100,000 resamplings was employed to simulate the effects of the sex ratio on the population parameters of *P. xylostella* and to estimate the standard error (SE). In addition, output files of bootstrap studies on the population parameters of *P. xylostella* relative to reproduction and population growth were used to compare the differences between treatments through one-way analysis of variance (ANOVA), and multiple comparisons among treatments were performed using the Student–Newman–Keuls (SNK) test. Meanwhile, the antioxidative enzyme activity and MDA levels in *P. xylostella* were calculated using Excel software (version 2010; Microsoft, Redmond, WA, USA). Similarly, one-way ANOVA was employed to compare the differences between HVEF exposure durations, and multiple comparisons of enzyme activity were made using the SNK test. All analyses were performed using SPSS 26.0 software (SPSS Inc., Chicago, IL, USA). The level of significance was set to $p$-value < 0.05. All graphs were prepared using GraphPad Prism 8.0 software (GraphPad Software, San Diego, CA, USA).

## 3. Results

### 3.1. HVEF Treatment Adversely Affected Net Reproduction Rate ($R_0$) of P. xylostella

Direct exposure of *P. xylostella* eggs to HVEF at 5.0 kV/cm adversely affected $R_0$, and the adverse effects exhibited different profiles at different time points. In particular, for the first generation of *P. xylostella*, although no significant difference in $R_0$ was detected when compared to the control group under HVEF stress ($p > 0.05$), HVEF treatment resulted in a reduced $R_0$ value, with the lowest value observed for the 5 min treatment (Figure 2A). When the stress continued to the third generation, $R_0$ decreased significantly with treatment times other than 5 min compared to the control, with the most significant decrease observed when the treatment time was 20 min ($p < 0.05$). In the fifth generation, 5 and 10 min treatments resulted in reduced $R_0$, and 10 min treatment resulted in significantly decreased $R_0$, whereas the $R_0$ values for 15 and 20 min treatments were close to the controls. In addition, under controlled greenhouse conditions, the presence of HVEF stress resulted in a significantly reduced $R_0$ in the third generation, especially with 10, 15, and 20 min treatments, while the $R_0$ values for fifth-generation *P. xylostella* showed an increasing trend with 15 and 20 min treatments (Figure 2B).

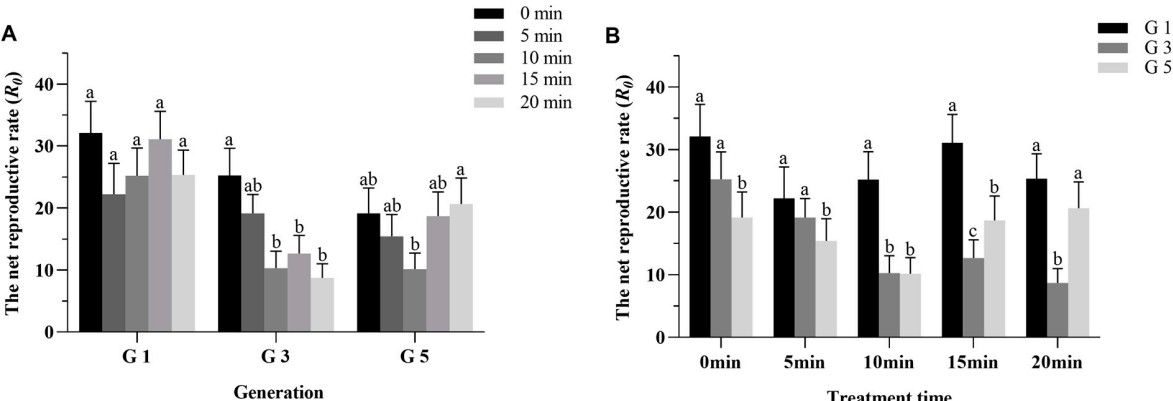

**Figure 2.** Net reproductive rate ($R_0$) of the first, third, and fifth generations of *P. xylostella* under HVEF stress. Note: Values are expressed as mean ± standard error (SE). (**A**) The different lowercase letters indicates a significant difference at $p < 0.05$ between generations for groups exposed to electrostatic fields (0, 5, 10, 15, and 20 min). (**B**) The different lowercase letters indicates significant differences at $p < 0.05$ between treatments for 3 generations (G1, G3, and G5). Differences were compared using the paired bootstrap test.

### 3.2. HVEF Stress Significantly Affected the r of P. xylostella in All Generations

To better evaluate the effect of direct exposure of *P. xylostella* to HVEF on population growth, we determined the value of *r*. The results showed that in the first generation, although *r* did not exhibit a significant decrease under HVEF treatment compared to the controls ($p > 0.05$), it showed significant differences between treatment durations ($p < 0.05$); the lowest and highest values were found for the 5 and 15 min treatment times, respectively (Figure 3A). In the third generation, except for the 5 min treatment group, *r* decreased significantly in the experimental groups compared to the control, and the most significant decrease was found with the 20 min treatment ($p < 0.05$). For the fifth generation, *r* showed a similar decreasing tendency to the first generation; there was a significant decrease between the 5 and 10 min treatments and the control, and the other treatments did not show a significant decrease. Regarding the multigenerational effects, *r* was significantly decreased at different treatment times, especially at 10, 15, and 20 min (Figure 3B). These results suggest that the reproductive capacity of the *P. xylostella* populations was affected by the reduced *r* and its derived parameters in the novel HVEF environment.

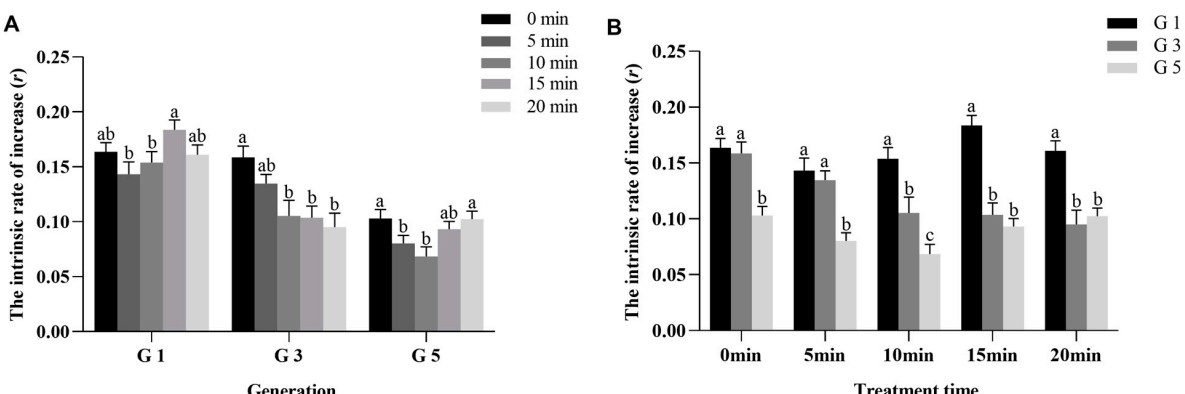

**Figure 3.** Intrinsic natural increase rate (*r*) of the first, third, and fifth generations of *P. xylostella* under HVEF stress. Note: Values are expressed as mean ± standard error (SE). (**A**) The different lowercase letters indicates a significant difference at $p < 0.05$ between generations for groups exposed to electrostatic fields (0, 5, 10, 15, and 20 min). (**B**) The different lowercase letters indicates significant differences at $p < 0.05$ between treatments for 3 generations (G1, G3, and G5). Differences were compared using the paired bootstrap test.

### 3.3. HVEF Treatment Gradually Prolonged the T of P. xylostella

In the first generation, the *T* of *P. xylostella* was decreased after HVEF exposure, with significant decreases at 15 and 20 min (Figure 4A). However, in the third and fifth generations, *T* was significantly increased by HVEF stress compared to the control ($p < 0.05$). In particular, the *T* values of *P. xylostella* in the third generation showed an increasing trend at all treatment times, with the most significant increase at 15 min ($p < 0.05$), and a significant increasing trend was also observed at 5 and 10 min in the fifth generation (Figure 4A). Regarding the effects of HVEF on multiple generations of *P. xylostella*, HVEF stress significantly prolonged *T* as the number of generations increased; the longest *T* was found for the 5 min treatment in the fifth generation (Figure 4B).

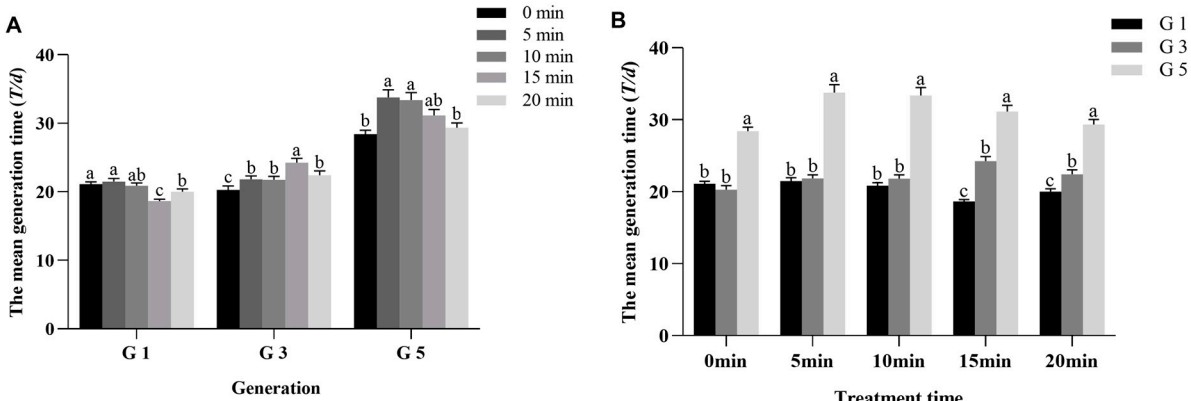

**Figure 4.** Mean generation time (*T*) of the first, third, and fifth generations of *P. xylostella* eggs in 5 successive generations after HVEF treatment. Values are expressed as mean ± standard error (SE). (**A**) The different lowercase letters indicates a significant difference at $p < 0.05$ between generations for groups exposed to electrostatic fields (0, 5, 10, 15, and 20 min). (**B**) The different lowercase letters indicates significant differences at $p < 0.05$ between treatments for 3 generations (G1, G3, and G5). Differences were compared using the paired bootstrap test.

### 3.4. HVEF Stress Exerted a Suppressive Effect on Population Growth of P. xylostella

As shown in Table 1, similarly to the tendency of *r* described above, the value of λ for *P. xylostella* was significantly reduced under HVEF treatment. In the first generation, there was no significant difference between HVEF exposure and the control group ($p > 0.05$). Compared with the control group, the value of λ in third-generation *P. xylostella* was significantly decreased after HVEF exposure, with the lowest value observed at 20 min. In the fifth generation, the value of λ showed a significant decrease with shorter 5 and 10 min treatment times ($p < 0.05$), and no significant decrease was observed with longer times. In addition, the value of λ in each treatment was the greatest for the first generation of *P. xylostella*, and the lowest for the fifth generation.

In comparison with the control, in the first generation, the population doubling time decreased significantly in the 15 min treatment group ($p < 0.05$), while the other treatment groups were not significantly affected by HVEF stress ($p > 0.05$). In the third generation, the population doubling time increased significantly with each treatment duration ($p < 0.05$), and the largest value of *dt* was observed for the 20 min treatment group. For the fifth generation, the largest value was found with the 10 min treatment time, and was significantly increased compared to the control ($p < 0.05$). When considering the multigenerational effects, the population doubling time of *P. xylostella* significantly increased as the number of generations increased, and direct exposure of eggs to HVEF at 5.0 kV/cm for 10 min resulted in the greatest increase in *dt* in the fifth generation. These results suggest that 10 min of HVEF exposure exerted a more suppressive effect on population growth.

**Table 1.** $\lambda$ and *dt* parameters of *P. xylostella* for each generation under HVEF at an intensity of 5.0 kV/cm for different durations.

| Statistic | Generation | 0 min | 5 min | 10 min | 15 min | 20 min |
|---|---|---|---|---|---|---|
| $\lambda$ | G1 | 1.180 ± 0.006 bXY | 1.164 ± 0.009 bX | 1.192 ± 0.011 aX | 1.223 ± 0.004 abX | 1.190 ± 0.013 abX |
| | G3 | 1.208 ± 0.011 aX | 1.178 ± 0.001 bX | 1.160 ± 0.009 bY | 1.155 ± 0.007 bY | 1.121 ± 0.009 cY |
| | G5 | 1.156 ± 0.010 aY | 1.104 ± 0.003 bY | 1.092 ± 0.005 bZ | 1.137 ± 0.008 aY | 1.139 ± 0.008 aY |
| *dt* | G1 | 4.191 ± 0.116 abXY | 4.600 ± 0.237 aY | 3.974 ± 0.212 abY | 3.450 ± 0.053 bY | 4.025 ± 0.280 abY |
| | G3 | 3.686 ± 0.167 cY | 4.230 ± 0.024 bcY | 4.706 ± 0.236 bY | 4.832 ± 0.201 bX | 6.101 ± 0.388 aX |
| | G5 | 4.819 ± 0.284 bX | 7.014 ± 0.196 aX | 7.944 ± 0.417 aX | 5.415 ± 0.286 bX | 5.341 ± 0.277 bX |

Note: $\lambda$ and *dt* represent the finite rate of increase and population doubling time, respectively. Values are expressed as mean ± standard error (SE). SE was estimated by using the bootstrap technique with 100,000 resamplings. Means followed by letters a–c in the same row are significantly different between treatment times in the same generation, according to paired bootstrap tests based on confidence intervals of differences at the 5% significance level, while letters X–Z indicate significant differences between generations (G1, G3, and G5) for the same HVEF duration.

*3.5. Effect of HVEF Exposure on Survival and Oviposition Parameters of P. xylostella*

To more accurately describe the survival probability among individuals, the age–stage survival rate of *P. xylostella* from egg to adult was investigated. The $l_x$ curve showed that the survival duration of *P. xylostella* adults increased as the number of generations increased, and the longest life span was detected in the fifth generation (Figure 5). Under HVEF stress, although the survival rate of each treatment group was higher than that of the control group, the survival rate rapidly declined compared with the control group in the first generation, with the fastest decrease observed for the 20 min treatment time. In addition, the life spans of females were significantly shorter than those of males, and females who received 15 min of treatment had the shortest life spans. With continued HVEF exposure, the survival rates of third-generation *P. xylostella* in the larval and pupal stages were still higher than that those of the control group, and declined rapidly at the pupal and adult stages with 10 min of treatment. In the fifth generation, the survival rate in the pupal stage after HVEF exposure was lower than that of the control group, and it declined evenly and slowly. Meanwhile, the survival times were longer than those of the first and third generations. Notably, the difference in survival rates between females and males in the first and third generations was not significant; however, in the fifth generation, the survival rates of the females were significantly higher than those of the males (Figure 5).

In addition, direct exposure to HVEF had an adverse effect on the adult preoviposition period (APOP) in *P. xylostella*, and the APOP gradually extended in the same generation with increased exposure time. As shown in Table 2, the APOP under 15 min of treatment was significantly longer than the control and the other treatment groups in all generations, except for the fifth. In the fifth generation, the APOP with 15 min of HVEF exposure decreased suddenly compared to other treatments, and was close to that of the control group, while the APOP at 10 min was significantly higher than in the other groups ($p < 0.05$). Although the oviposition period (OP) and oviposition days (OD) of female *P. xylostella* were not significantly affected by direct HVEF exposure, they showed an increasing trend in the generations, with the longest OP and OD detected in the fifth generation after all treatments. The shortest OD and OP were observed with 10 min of treatment. Moreover, although this was not observed in the first generation, HVEF exposure resulted in a decreased number of eggs per day during oviposition in the third and fifth generations compared to the controls, with the lowest number found after 10 min of treatment in all test generations. The fecundity of the females showed a similar trend to the number of eggs per day during oviposition in each treatment group over multiple generations. These results suggest that direct HVEF exposure has an adverse effect on the survival and oviposition of *P. xylostella*.

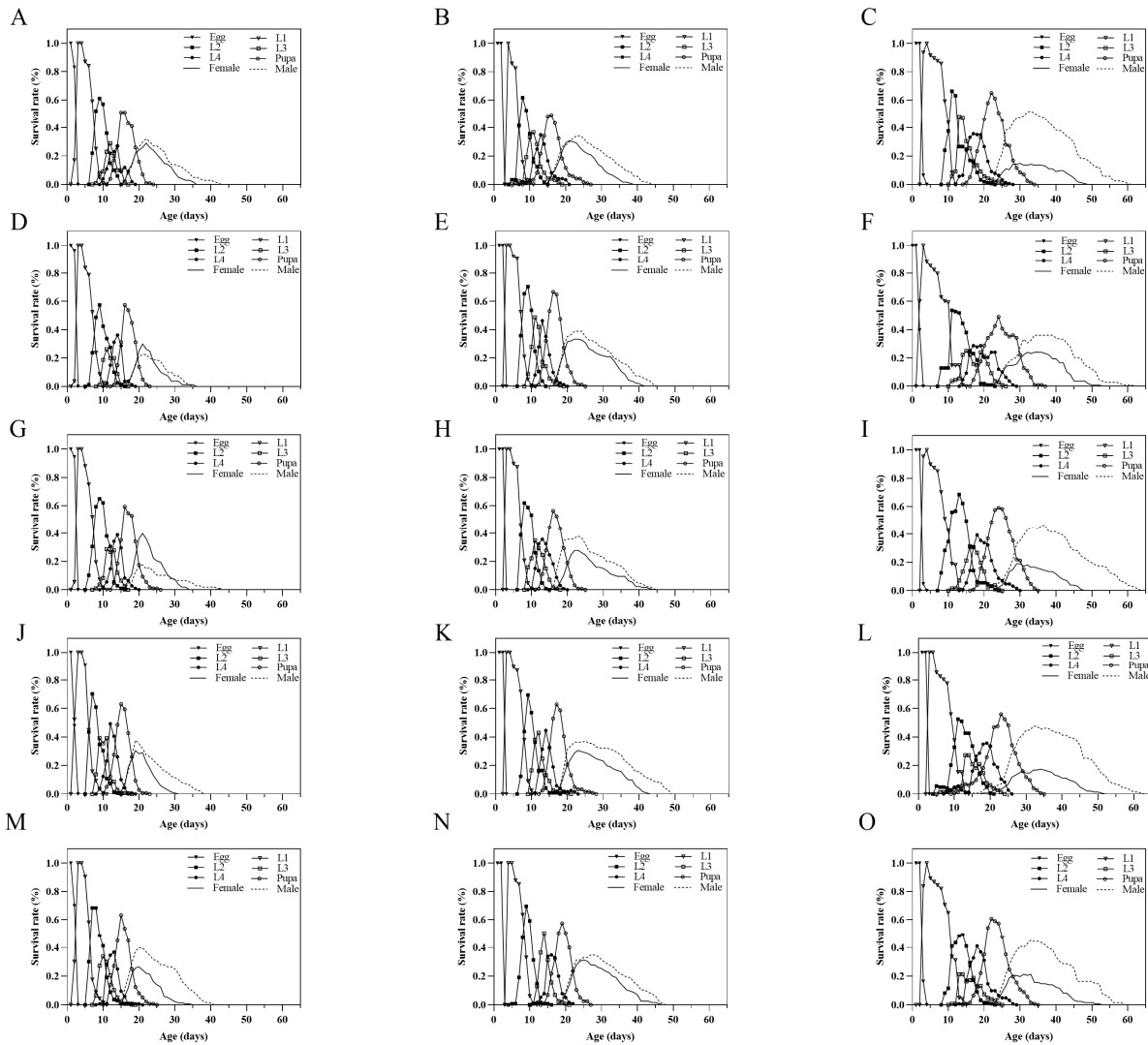

**Figure 5.** Age-specific survival rates ($l_x$) at each life stage of the first, third, and fifth generations of *P. xylostella*, with 5 consecutive generations treated with HVEF at 5.0 kV/cm for 0, 5, 10, 15, and 20 min. For each generation, the control group received 0 min of treatment. (**A,D,G,J,M**) The first column represents the survival rate of each stage at 0, 5, 10, 15, and 20 min for the first generation. (**B,E,H,K,N**) The second column represents the survival rate with each treatment time for the third generation. (**C,F,I,L,O**) The last column represents the survival rate of the fifth generation after HVEF stress at each stage.

**Table 2.** Fecundity parameters of each generation under different treatment times.

| Generation | Treatment Time (min) | Adult Preoviposition Period (APOP) (d) | Oviposition Period (OP) (d) | Oviposition Days (OD) (d) | Eggs per Day during Oviposition Period | Fecundity per Female |
|---|---|---|---|---|---|---|
| | 0 | 0.77 ± 0.184 aX | 5.66 ± 0.502 aY | 5.23 ± 0.440 aX | 18.866 ± 1.680 abX | 91.714 ± 7.650 aX |
| | 5 | 1.19 ± 0.184 aX | 4.83 ± 0.557 aZ | 4.25 ± 0.501 aY | 16.550 ± 2.272 abX | 74.125 ± 10.856 abX |
| G1 | 10 | 1.13 ± 0.229 aY | 4.30 ± 0.379 aY | 3.83 ± 0.335 aY | 13.339 ± 1.715 bX | 57.957 ± 8.162 bX |
| | 15 | 1.24 ± 0.297 aY | 4.51 ± 0.330 aZ | 4.09 ± 0.291 aY | 22.432 ± 2.099 aX | 95.378 ± 7.606 aX |
| | 20 | 1.04 ± 0.217 aX | 5.12 ± 0.435 aY | 5.23 ± 0.440 aX | 18.145 ± 1.613 abX | 84.049 ± 7.580 abX |
| | 0 | 1.42 ± 0.433 aX | 6.36 ± 0.526 aY | 5.28 ± 0.457 aX | 13.504 ± 1.281 aY | 84.167 ± 8.912 aX |
| | 5 | 2.09 ± 0.401 aX | 7.84 ± 0.709 aY | 5.98 ± 0.504 aXY | 10.156 ± 1.141 abY | 68.911 ± 6.734 abX |
| G3 | 10 | 2.22 ± 0.462 aXY | 5.70 ± 0.658 aY | 4.91 ± 0.579 aXY | 9.140 ± 1.329 abX | 56.348 ± 10.946 abX |
| | 15 | 2.52 ± 0.862 aX | 7.52 ± 0.996 aY | 5.87 ± 0.695 aX | 11.087 ± 2.087 abY | 66.565 ± 9.159 abX |
| | 20 | 1.82 ± 0.440 aX | 5.73 ± 0.715 aXY | 4.64 ± 0.605 aX | 7.608 ± 0.885 bZ | 45.591 ± 8.294 bY |

**Table 2.** *Cont.*

| Generation | Treatment Time (min) | Adult Preoviposition Period (APOP) (d) | Oviposition Period (OP) (d) | Oviposition Days (OD) (d) | Eggs per Day during Oviposition Period | Fecundity per Female |
| --- | --- | --- | --- | --- | --- | --- |
| | 0 | 0.25 ± 0.083 bX | 8.36 ± 0.853 aX | 6.57 ± 0.791 aX | 13.253 ± 1.609 aY | 105.143 ± 13.774 aX |
| | 5 | 1.33 ± 0.211 bX | 10.52 ± 1.133 aX | 7.24 ± 0.828 aX | 8.446 ± 0.857 aY | 79.476 ± 9.600 aX |
| G5 | 10 | 2.95 ± 0.671 aX | 7.50 ± 0.947 aX | 5.70 ± 0.758 aX | 8.688 ± 1.256 aX | 64.500 ± 9.876 aX |
| | 15 | 0.60 ± 0.212 bY | 9.77 ± 0.956 aX | 7.27 ± 0.733 aX | 9.467 ± 1.369 aY | 89.000 ± 12.100 aX |
| | 20 | 1.55 ± 0.652 bX | 7.4 ± 0.868 aX | 6.21 ± 0.609 aX | 12.240 ± 1.280 aY | 87.000 ± 10.787 aX |

Note: Values are expressed as mean ± standard error (SE). SE was estimated using the bootstrap technique with 100,000 resamplings. Means followed by letters a–c in the same row were significantly different between treatment times in the same generation according to the paired bootstrap test, which is based on a confidence interval of differences at the 5% significance level, while letters X–Z indicate significant differences between generations (G1, G3, and G5) for the same HVEF duration.

### 3.6. HVEF Exposure Led to Significant Oxidative Damage to Fourth-Instar P. xylostella

Based on one-way ANOVA, HVEF stress significantly affected the antioxidative enzyme activity, subsequently increasing the MDA level of instar *P. xylostella*. In particular, total SOD activity exhibited an increasing trend after HVEF treatment (G1: $F$ = 16.171, *d.f.* = 4, $p < 0.05$; G3: $F$ = 38.938, *d.f.* = 4, $p < 0.05$), except for the 20 min treatment in the third generation, when compared with the controls (Figure 6A). The first and third generations exhibited the greatest increase in total SOD activity at 10 min. Similarly, when compared with the controls, the CAT (G1: $F$ = 29.482, *d.f.* = 4, $p < 0.05$; G3: $F$ = 35.861, *d.f.* = 4, $p < 0.05$) and POD (G1: $F$ = 85.530, *d.f.* = 4, $p < 0.05$; G3: $F$ = 27.839, *d.f.* = 4, $p < 0.05$) activity under most treatment durations showed an increasing trend, while a suppressive effect on POD activity was found with other treatment durations, including 15 min in the first generation and 10 min in the third generation, and the lowest POD activity in each generation was found with these durations (Figure 6B,C). Moreover, alterations in the activity of these antioxidative enzymes induced a significantly increased MDA level in fourth-instar *P. xylostella* (G1: $F$ = 3.988, *d.f.* = 4, $p < 0.05$; G3: $F$ = 5.521, *d.f.* = 4, $p < 0.05$). The highest MDA level was found after 20 min in the first generation and after 10 min in the third generation, while the controls exhibited the lowest MDA levels (Figure 6D).

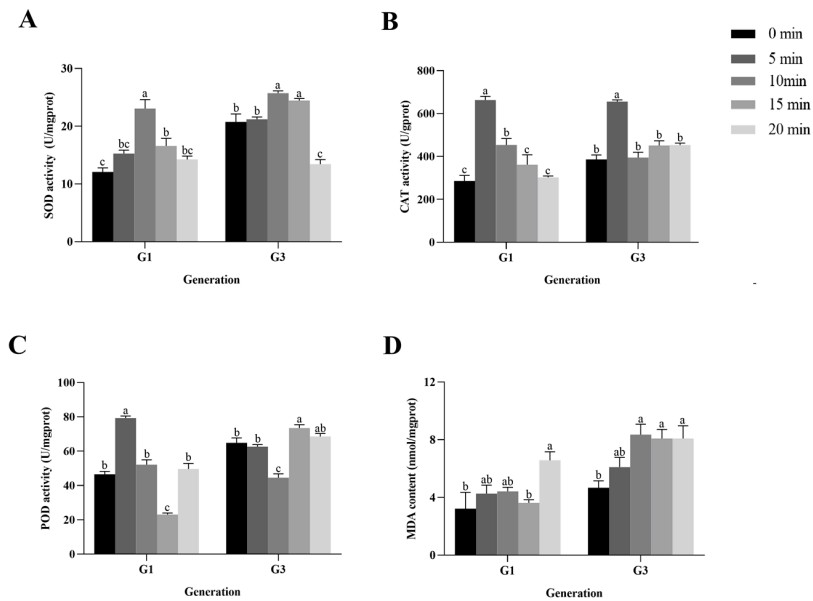

**Figure 6.** Effects of different durations of HVEF exposure on antioxidative enzyme activity and malondialdehyde levels of fourth-instar *P. xylostella*. Values are expressed as mean ± standard error (SE). (**A**) Superoxide dismutase (SOD), (**B**) catalase (CAT), (**C**) peroxidase (POD), (**D**) malondialdehyde (MDA). Different lowercase letters indicate significant differences ($p < 0.05$).

## 4. Discussion

In the present study, direct exposure of *P. xylostella* eggs to HVEF resulted in oxidative damage and adversely affected parameters associated with growth, and development, and reproduction in multiple consecutive generations. Interestingly, the 10 min treatment time had the most significant effect on all studied parameters when compared with the controls. These results were consistent with our previous study on direct and indirect exposure of first-instar cereal aphid nymphs to HVEF [19,28,29]. For instance, the results showed that exposure of either cereal aphids or wheat seeds to HVEF at 4 kV/cm for 20 min significantly affected the growth and reproductive parameters of aphid populations, and that the antioxidant enzyme activity was altered under stress [21,28,29]. In addition, in studies of other types of abiotic stress, we found that treating *P. xylostella* at different temperatures had a reproductive impact, with further evidence of reproductive compensation [30,31]. Furthermore, direct exposure of *P. xylostella* to $^{60}$Co-γ radiation resulted in significantly reduced nymphal fecundity and postfecundity, as well as reduced adult longevity and fecundity, suggesting that either HVEF or other abiotic stresses adversely affect the performance of *P. xylostella* [32,33].

Previous studies have demonstrated that exposing cereal aphids to HVEF immediately affected their performance, and that they gradually exhibited adaptability to electric field stress when the treatment was continued for 10 more generations [19,29,34]. Interestingly, the growth and development, fecundity, and population growth of *P. xylostella* were not immediately inhibited in the first generation; when the treatment was continued up to the fifth generation, more serious adverse effects on their performance under electric field stress were observed. This is probably because the external eggshells of *P. xylostella* eggs can partially suppress the adverse effects of HVEF exposure, and as a result, some of the population dynamics parameters did not rapidly exhibit significant effects. Meanwhile, some of the parameters, such as *r*, could be determined by $R_0$ and *T*; a higher value of $R_0$ and a lower *T* can lead to a higher value of *r* in the first generation than subsequent generations. The curve of the age–stage survival rate of *P. xylostella* supports this. For instance, direct HVEF exposure rapidly reduced the survival rate of *P. xylostella* in the first and third generations; when the treatment was continued up to the fifth generation, they adapted to HVEF, which eventually allowed them to survive and persist under electric field stress. Accordingly, the growth cycle of *P. xylostella* is longer than that of cereal aphids, and they recover rapidly from the adverse effects of HVEF exposure. Moreover, it is well known that when the environment is suitable, aphids generally undergo parthenogenesis, while *P. xylostella* is oviparous, producing eggs [35]. Thus, more work is required to unravel the exact nature of the damage to *P. xylostella* that is caused by HVEF exposure.

In addition, the current study suggests that the dose-dependent effects of different intensities of HVEF exposure on *P. xylostella* eggs produce different stress effects. Previous studies have shown that electric or magnetic fields of moderate intensity have the greatest inhibitory effect on the biological performance of different species of organisms [36–38]. For instance, static magnetic field (SMF) exposure at a moderate intensity of 0.2–0.4 T was found to maximally affect leukemia cell proliferation and the cell cycle [39]. Meanwhile, by treating rats with static magnetic fields of different intensities, it was demonstrated that static magnetic fields of moderate intensity inhibited osteocalcin secretion and human osteoblast-like cell proliferation [40]. The same results were reported in a study which attempted to enhance sorghum seed viability by using HVEF [41].

In this study, the life table parameters $R_0$ and *r* of the first generation of *P. xylostella* subjected to HVEF stress decreased significantly under moderate treatment durations, but other parameters were not significantly altered compared to the control group. It is possible that electrostatic damage is repaired within the organism by its own defense system, or that the population is maintained by reducing reproductive capacity. Nevertheless, as the number of generations exposed to HVEF treatment increased, the damage caused by HVEF gradually accumulated and reached a threshold where it could be repaired by the moths themselves. Therefore, in the third and fifth generations, there were significant changes in

vital phenotypes compared with the first generation; for example, the population doubling time increased significantly with increased treatment generations, but between generations, it showed a dose effect, i.e., longer population doubling times with shorter treatment durations. In contrast, both reproduction-related parameters, $R_0$ and $r$, were significantly lower in the 10 min treatment group than in the control. This proves that different radiation durations can have different toxic effects on insects, but not that a longer duration will produce a more pronounced effect.

In agreement with previous studies, the current study suggests that direct exposure of organisms to different environmental stressors, such as heavy metals, UV radiation, or HVEF, could cause the production of large amounts of reactive oxygen species (ROS). ROS cause an imbalance in the normal oxygen-consuming metabolic processes in the body, and have toxic effects; for example, they cause changes in cell structure and protein function, even causing structural changes in DNA, and, thus, its function [42–45]. For instance, direct exposure of *P. xylostella* eggs to HVEF for 10 min resulted in the greatest oxidative stress and the most significant adverse effects on performance. A similar result was found in *Thitarodes xiaojinensis* (Lepidoptera: Hepialidae) after using heat stress, which indicates that a large number of oxygen radicals were produced. This caused structural and functional damage to mitochondria, resulting in a series of cellular dysfunctions which led to cell and tissue death, and, ultimately, to irreversible damage to the insects, causing them to live longer [46].

In response to various adverse environmental changes, insects have evolved complex antioxidant enzyme protection mechanisms to mitigate the harmful effects of oxidative damage [46–49]. SOD, CAT, and POD play important roles in eliminating extra ROS compounds [50,51], and changes in their gene expression levels or enzyme activity can reflect the state of environmental stress and the degree of oxidative damage to the organism. For example, treating *P. xylostella* eggs with $^{60}$Co-$\gamma$ radiation caused significant changes in SOD, CAT, and POD gene expression and enzyme activity [51,52]. Apart from that, the results showed that *P. xylostella* midgut microbes encode for large amounts of SOD, CAT, and POD, which helps the host to reduce ROS to nontoxic compounds [53]. Although symbiotic bacteria are also present in aphids, they have completely different roles; for example, the obligate species *Buchnera aphidicola* can transform nonessential amino acids into amino acids that aphids cannot synthesize, and some species of secondary endosymbionts provide energy materials and some detoxification functions. This might be an important reason why HVEF exposure was shown to have a stronger adverse effect on aphids than *P. xylostella* [29,54,55].

In addition, direct exposure of *P. xylostella* eggs to $^{60}$Co-$\gamma$ radiation at 200 Gy resulted in significantly upregulated expression of the heat-stimulated protein (HSP) 70 genes [51]. It was demonstrated that some species of HSP have antioxidant capacity, which means that they can inhibit or scavenge the excess free radicals produced by organisms. Exposure to different external adverse environmental stressors (such as high temperature, low temperature, and UV radiation) impairs regular protein function and disrupts cellular homeostasis. Thereafter, large numbers of HSPs are expressed to help organisms to withstand external stresses and to enhance their resilience [53,56,57]. UV-A irradiation of *Ostrinia furnacalis* resulted in a slow decrease, then a rapid increase, and then a sudden decrease in the expression of vitellogenin receptor (*VgR*), a gene related to reproduction, and a corresponding change in fertility occurred in response to the effects of the external environment [58]. Furthermore, in the present study, a significant increase in malondialdehyde (MDA) content was detected after HVEF exposure, which may be a crucial marker of oxidative damage. This may be because MDA can easily bind to proteins and DNA and damage the structure and function of biomolecules [48,59]. Therefore, in subsequent experiments, we can further explore the biological roles of important proteins, such as HSP and reproduction-related proteins, in the adaptation of organisms to new HVEF environments.

In summary, direct exposure of *P. xylostella* eggs to HVEF at an intensity of 5.0 kV/cm for 10 min had the strongest inhibition effects on their performance, possibly resulting in longer growth cycles and increased population doubling time, as well as reduced

reproductive capacity. The results of this study will help us to understand the effects of HVEF stress on the performance of *P. xylostella* and its adaptation mechanisms, and will provide experimental data and a theoretical basis for control strategies in the new HVEF environment (Figure 7).

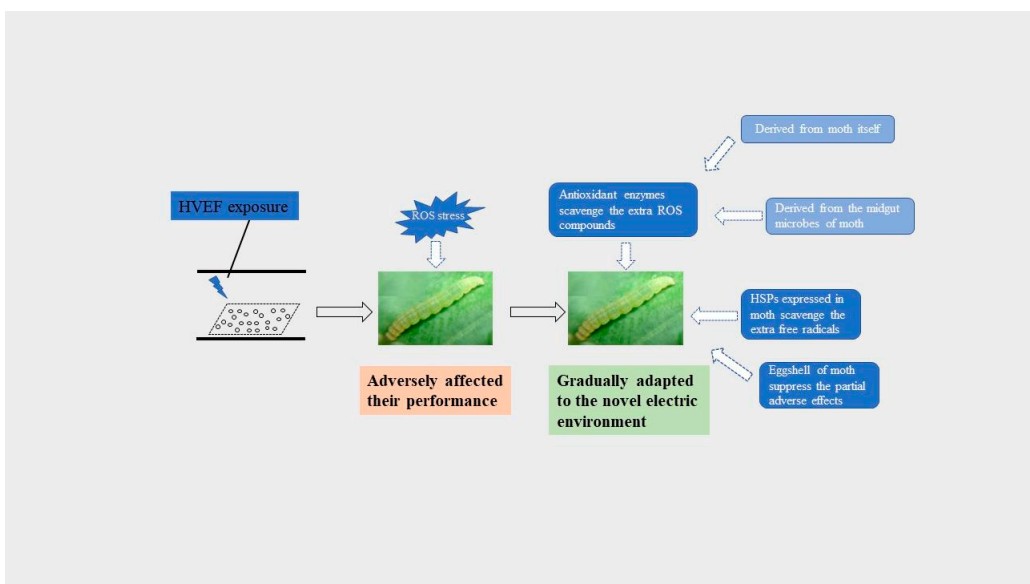

**Figure 7.** Proposed alterations in *P. xylostella* after exposure to HVEF stress based on findings obtained in or deduced from this study. Solid arrows represent pathways supported by experimental evidence from the present study; dotted arrows represent potential physiological alterations predicted from the literature.

**Author Contributions:** L.J.: investigation, writing—original draft, data curation. S.X.: conceptualization, data curation, writing—review and editing. H.S.: writing—original draft, data curation, investigation. J.G.: writing—original draft, investigation. X.Y.: investigation, data curation. C.L.: writing—review and editing, data curation. G.L.: writing—review and editing, data curation. K.L.: conceptualization, writing—review and editing, supervision, data curation, funding acquisition. All authors have read and agreed to the published version of the manuscript.

**Funding:** This work was supported by the Research Fund for the Doctoral Starting up Foundation of Yan'an University, China (No. YDBK2019-65), the Educational Innovation Program of Graduate Students of Yan'an University (YCX2021075), and the Innovation and Entrepreneurship Training Program for College Student of Yan'an University (D2020095).

**Institutional Review Board Statement:** Not applicable.

**Informed Consent Statement:** Not applicable.

**Data Availability Statement:** The data presented in this study are available on request from the corresponding authors.

**Conflicts of Interest:** The authors have no conflict of interest to declare.

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
