# Peer review of "High-Voltage Electrostatic Fields Adversely Affect the Performance of Diamondback Moths over Five Consecutive Generations"

_agronomy, doi:10.3390/agronomy13041008_

Round 1

Reviewer 1 Report

The paper by Jia et al. presents the results of an intriguing study examining the effects of high-voltage electrostatic fields on the development, reproduction, etc. of the diamondback moth, Plutella xylostella. While the results do not contribute much to the control of this pest, they are nonetheless very interesting because the similar EMF has become an environmental concern in places where powerlines are concentrated. In these places, there is always a question of how EMFs are effecting organisms that live within the influence of these fields.

The methods and results of the study are detailed and compelling.

The greatest shortcoming of the paper is the language, which requires considerable editing. When I accepted the manuscript for review, I indicated that I would review it only if I could obtain a copy in Word, which I did not receive. Hence, although I am willing to edit the language, I was not given the opportunity to do so.

Once the language is corrected, I think this paper will make an interesting and rather novel contribution.

Author Response

Dear Editor and Reviewers,

Thank you very much for your letter and the comments about our paper submitted to your journal. We would also like to thank the reviewers for their careful and constructive reviews. Soon after receiving your comments, the other authors and I carefully revised the manuscript. The changes made according to the comments from the reviewers are detailed below.

Reviewer #1:

The paper by Jia et al. presents the results of an intriguing study examining the effects of high-voltage electrostatic fields on the development, reproduction, etc. of the diamondback moth, Plutella xylostella. While the results do not contribute much to the control of this pest, they are nonetheless very interesting because the similar EMF has become an environmental concern in places where powerlines are concentrated. In these places, there is always a question of how EMFs are effecting organisms that live within the influence of these fields. The methods and results of the study are detailed and compelling.

Answer: Thank you very much for carefully reviewing our manuscript and kindly give us many suggestions and comments for improving the readability of this manuscript. Yes, this study focused more on characterizing the performance of diamondback moths to the changing electrical environment. It looks like the findings of this study contribute less to the control of this pest, however, it provides experimental data and a theoretical basis for the development of more effective and sustainable pest management strategy to control P. xylostella larvae. In addition, for improving the readability and logicality of texts in this manuscript, we carefully modified the words and corrected the illogical points in the abstract, introduction, results, discussion sections, and added more content to explain our results. For the tables and figures, we carefully checked the data analysis of this study, and supplied some missing information in tables, and formatted some figures in a corrected manner. Please allow us to make those revisions.

The greatest shortcoming of the paper is the language, which requires considerable editing. When I accepted the manuscript for review, I indicated that I would review it only if I could obtain a copy in Word, which I did not receive. Hence, although I am willing to edit the language, I was not given the opportunity to do so. Once the language is corrected, I think this paper will make an interesting and rather novel contribution.

Answer: Yes, the quality of English language in original submission is not quite well. In the revision, we had requested the English editing services in MDPI to improve the English and the readability of this manuscript. We guess the text in the revision properly described the content of this manuscript to readers.

Reviewer 2 Report

Interesting innovative work that is could be important that could be important for the field of IPM. The results are interesting, but the results so far appear to be somewhat inconclusive, and usage in IPM still unclear. The data could be more clearly represented. there is a wealth of data, however following it is difficult. Also, the font sizes vary and english needs some correction. I appreciate this is innovative and the data important, however streamlining of the imformation presented with clearer conclusions would be useful. The data is muddy and the error bars over lap, in places where "effect" was noted in the abstract. I woudl like to this again, but as is I can't recommend for publication. 

line 13/14: "..., two-sex life table over..." please rephrase highlighted section. First time mentioned should be explained what this is - not the second time. 

Line 14: The cohort eggs of P. xylostella - suggest rephrasing, for correct grammar. 

Line 26: "The present findings will provide experimental evidence" suggest deleting highlighted word. It would be better to have some indication of cost effectiveness and hypothesized application method in the abstract provided. 

Please review your data/ results section, your statistics including significance of data, particularly in relation to conclusions made and the statements in the abstract. 

Author Response

Dear Editor and Reviewers,

Thank you very much for your letter and the comments about our paper submitted to your journal. We would also like to thank the reviewers for their careful and constructive reviews. Soon after receiving your comments, the other authors and I carefully revised the manuscript. The changes made according to the comments from the reviewers are detailed below.

Reviewer #2:

Interesting innovative work that is could be important that could be important for the field of IPM. The results are interesting, but the results so far appear to be somewhat inconclusive, and usage in IPM still unclear. The data could be more clearly represented. there is a wealth of data, however following it is difficult. Also, the font sizes vary and english needs some correction. I appreciate this is innovative and the data important, however streamlining of the imformation presented with clearer conclusions would be useful. The data is muddy and the error bars over lap, in places where "effect" was noted in the abstract. I would like to this again, but as is I can't recommend for publication.

Answer: Thank you very much for carefully reviewing our manuscript and kindly giving us many suggestions and comments for improving the readability of this manuscript. In the revision, we carefully followed your corrections. This study focused more on characterizing the performance of diamondback moths to the changing electrical environment through the determination of the population parameters of P. xylostella. In addition, we determined the biochemical basis of HVEF exposure on P. xylostella through evaluation its superoxide dismutase activity. It looks like the findings of this study contribute less to the control of this pest, however, it provides experimental data and a theoretical basis for the development of more effective and sustainable pest management strategy to control P. xylostella larvae. For more accuracy illustrating the results and conclusions of this study, we carefully checked the data analysis of this study, and supplied some missing information in tables, and formatted some figures in a corrected manner. Because the TWOSEX-MSChart software was employed to determine the population parameters for all P. xylostella individuals in the study. For R0 parameter, its standard errors always reach for 1/3 of its value [Wei M, Chi H, Guo Y, Li X, Zhao L, Ma R. Demography of Cacopsylla chinensis (Hemiptera: Psyllidae) Reared on Four Cultivars of Pyrus bretschneideri (Rosales: Rosaceae) and P. communis Pears With Estimations of Confidence Intervals of Specific Life Table Statistics. J Econ Entomol 2020, 113(5): 2343-2353.], which estimated by using bootstrap technique with 100,000 resamplings. Therefore, the values in some cases are less than error bars. In the revision, we carefully improving the logic and readability of abstract, and provide more "effect" with data. Moreover, we had requested the English editing services in MDPI to improve the English and the readability of this manuscript. We guess the text in the revision properly described the content of this manuscript to readers.

line 13/14: "..., two-sex life table over..." please rephrase highlighted section. First time mentioned should be explained what this is - not the second time.

Answer: Thank you very much for the rephrase suggestion. Sorry for we didn’t prepare a qualified abstracts in the original submission. In the revision, we added a clear objective and a concise enough summary in the main results of this study in the abstract. For improving the logic and readability of abstract, we deleted some unrelated sentences and reworded some sentences. Please allow us to make those revisions. Actually, the age–stage, two-sex life table is the common methods for analysis of insects population dynamitics, it is the correct phrase. Therefore, we didn’t rephrase this method.

Line 14: The cohort eggs of P. xylostella - suggest rephrasing, for correct grammar. 

Answer: Thank you very much for the rephrase suggestion. In the revision, we had replaced “the cohort eggs” with “the age-cohort eggs” from that sentence in the revision.

Line 26: "The present findings will provide experimental evidence" suggest deleting highlighted word. It would be better to have some indication of cost effectiveness and hypothesized application method in the abstract provided.

Answer: Thank you very much for your comments and suggestions. We apologize for a qualified abstracts in the original submission. In the revision, to improve the readability and logic of the abstract, after careful consideration, we deleted some illogic sentences and added more information into the abstract section. In addition, as described previously, this study focused more on describing the fitness of moths on the novel electrical environment, which will provide experimental evidence and a theoretical basis for the development of more effective and sustainable pest management strategy to control P. xylostella larvae. For the limitation of length of abstract, we didn’t provide the detail about how to apply our findings into agricultural production in future. Please forgive us not follow your suggestion.

Please review your data/ results section, your statistics including significance of data, particularly in relation to conclusions made and the statements in the abstract.

Answer: As described previously, we carefully checked the data analysis of this study, and supplied some missing information in tables, and formatted some figures in a corrected manner. In addition, we rewritten or provided more details to illustrate our results. Please check it in the abstract and results section and allow us to make those revisions.

Reviewer 3 Report

Reviewer report

For the manuscript draft of the article

High-Voltage Electrostatic Fields Adversely Affect the Performance of the Diamondback Moth over Five Consecutive Generations

The manuscript I have read is well written. Some note for the mention:

Figs 2 and 3 – the change in the factors in some cases are less than the value of statistical deviations in the compared options…

Despite of this authors have examined all results in good science way.

Author Response

Dear Editor and Reviewers,

Thank you very much for your letter and the comments about our paper submitted to your journal. We would also like to thank the reviewers for their careful and constructive reviews. Soon after receiving your comments, the other authors and I carefully revised the manuscript. The changes made according to the comments from the reviewers are detailed below.

Reviewer #3:

The manuscript I have read is well written. Some note for the mention:

Answer: Thank you very much for carefully reviewing our manuscript and kindly giving us many suggestions and comments for improving the readability of this manuscript. In the revision, we carefully followed your corrections.

Figs 2 and 3 – the change in the factors in some cases are less than the value of statistical deviations in the compared options…

Answer: For the tables and figures, we carefully checked the data analysis of this study, and supplied some missing information in tables, and formatted some figures in a corrected manner. Because the TWOSEX-MSChart software was employed to determine the population parameters for all P. xylostella individuals in the study. According to the instructions of TWOSEX-MSChart software, the standard errors of the population parameters of P. xylostella for each generation under HVEF at an intensity of 5.0 kV/cm for different durations were estimated by using bootstrap technique with 100,000 resamplings. Especially, for R0 parameter, its standard errors always reach for 1/3 of its value [Wei M, Chi H, Guo Y, Li X, Zhao L, Ma R. Demography of Cacopsylla chinensis (Hemiptera: Psyllidae) Reared on Four Cultivars of Pyrus bretschneideri (Rosales: Rosaceae) and P. communis Pears With Estimations of Confidence Intervals of Specific Life Table Statistics. J Econ Entomol 2020, 113(5): 2343-2353.]. Therefore, the changes in some cases are less than statistical deviations.

Despite of this authors have examined all results in good science way.

Answer: For improving the readability and logicality of texts in this manuscript, we carefully modified the words and corrected the illogical points in the abstract, results, discussion sections, and added more content to carefully illustrate and explain our results. Please allow us to make those revisions. In addition, we had requested the English editing services in MDPI to improve the English and the readability of this manuscript.